# AC Electric Conductivity of High Pressure and High Temperature Formed NaFePO_4_ Glassy Nanocomposite

**DOI:** 10.3390/nano14181492

**Published:** 2024-09-13

**Authors:** Aleksander Szpakiewicz-Szatan, Szymon Starzonek, Jerzy E. Garbarczyk, Tomasz K. Pietrzak, Michał Boćkowski, Sylwester J. Rzoska

**Affiliations:** 1Faculty of Physics, Warsaw University of Technology, 00-661 Warsaw, Poland; jerzy.garbarczyk@pw.edu.pl (J.E.G.); tomasz.pietrzak@pw.edu.pl (T.K.P.); 2Laboratory of Physics, Faculty of Electrical Engineering, University of Ljubljana, 1000 Ljubljana, Slovenia; starzonek@hotmail.com; 3Institute of High Pressure Physics of the Polish Academy of Sciences, 01-142 Warsaw, Poland; bocian@unipress.waw.pl

**Keywords:** high-pressure effects, alluaudites, NASICON, AC-conductivity, Jonscher scaling, sodium-based glassy batteries

## Abstract

Olivine-like NaFePO_4_ glasses and nanocomposites are promising materials for cathodes in sodium batteries. Our previous studies focused on the preparation of NaFePO_4_ glass, transforming it into a nanocomposite using high-pressure–high-temperature treatment, and comparing both materials’ structural, thermal, and DC electric conductivity. This work focuses on specific features of AC electric conductivity, containing messages on the dynamics of translational processes. Conductivity spectra measured at various temperatures are scaled by apparent DC conductivity and plotted against frequency scaled by DC conductivity and temperature in a so-called *master curve* representation. Both glass and nanocomposite conductivity spectra are used to test the (effective) exponent using Jonscher’s scaling law. In both materials, the values of exponent range from 0.3 to 0.9, with different relation to temperature. It corresponds to the electronic conduction mechanism change from low-temperature Mott’s variable range hopping (between Fe^2+^/Fe^3+^ centers) to phonon-assisted hopping, which was suggested by previous DC measurements. Following the pressure treatment, AC conductivity activation energies were reduced from $E_{AC}\approx 0.40\;eV$ for glass to $E_{AC}\approx 0.18\;eV$ for nanocomposite and are lower than their DC counterpart, following a typical empirical relation with the value of the exponent. While pressure treatment leads to a 2–3-orders-of-magnitude rise in the AC and apparent DC conductivity due to the reduced distance between the hopping centers, a nonmonotonic relation of AC power exponent and temperature is observed. It occurs due to the disturbance of polaron interactions with Na^+^ mobile ions.

## 1. Introduction

Modern energy storage is mainly based on lithium batteries [1,2,3]. Despite this element’s multiple advantages, such as low weight and wide range of cathode and anode materials suitable for various applications (i.e., small batteries for cell phones or large grid-level batteries), it has two main drawbacks: its limited availability and the environmental impact of its excavation and processing. This is why more sustainable alternative is required. Sodium is the first logical replacement for lithium due to its electro-chemical similarity and much higher abundance than lithium [4,5,6,7,8]. While poorer electrical and electrochemical properties limit the application of sodium batteries, they are still worth considering [9,10,11]. The selected material (NaFePO_4_-based glass and nanocomposite) is a sodium analog of LiFePO_4_ proposed by J. Goodenough in 1997 [7]. It contains only abundant, environmentally neutral elements (sodium, oxygen, iron, phosphorus) and presents a high theoretical gravimetric capacity (154 mAh/g) [8].

Glass materials have short-range, ordered structures that can adapt to significant lattice distortion, while notable specific surface areas may provide more storage sites than crystalline analog. Furthermore, the capacity of glass material can be controlled by modifying its composition. Those properties make amorphous solids (glasses) a viable replacement for crystalline materials when used as a cathode in new-generation batteries [12,13,14,15]. The main disadvantage of glass-based materials is their limited conductivity. The lab of the authors introduced an innovative way for the formation of a nanocomposite (crystallites with average grain size below 100 nm surrounded by glass matrix) using simultaneous high pressures and high temperature in the close vicinity of the glass temperature [15,16,17,18,19], and a 1–2-orders-of-magnitude conductivity improvement was observed in a glass containing lithium [16] and sodium [18].

The impact of high pressure on the physical and electrical properties of sodium-based nanomaterials is a subject of significant scientific inquiry, driven by its potential for various technological applications. Under high-pressure conditions, these nanomaterials undergo profound changes that affect their behavior at the atomic and electronic levels. A notable effect of high-pressure and high-temperature (HPHT) treatment is its ability to induce structural transformations in sodium-based nanomaterials. This pressure-induced structural change can lead to the formation of new crystal structures or alterations in existing ones. These structural modifications often result in changes in the material’s density (and thus volumetric capacity), lattice parameters, and atomic arrangements, which remain stable at ambient conditions [16,17,18,19,20,21] and often differ from other treatments [16,19,20,21].

Cathode materials should exhibit mixed electronic–ionic electric conduction (electric charge is simultaneously transferred by ions and electrons moving in opposite directions, contributing to total electric current), with the electronic one being predominant. In amorphous and disordered materials, electrons are transported by a hopping mechanism between transition metal centers (e.g., Fe^2+^/Fe^3+^ or V^4+^/V^5+^). It is expected that the temperature dependence of such hopping is described by Mott’s theory [21]. At a low-temperature range (below 1/4 of Debye temperature ${ϴ}_{D}$), variable range hopping (VRH) dominates. This means that “electrons are seeking” energetically close sites, but not necessarily neighboring ones. Electric conductivity in this temperature range is expressed by the following formula [18,19,22,23]:(1a)$$\sigma \left(T\right)=C\exp\left(\frac{-B}{T^{0.25}}\right)\Rightarrow ln\sigma \left(T\right)=lnC-BT^{\alpha =-1/4}$$
(1b)$$\frac{dln\sigma \left(T\right)}{dT}=\frac{1}{4}BT^{\alpha -1=-5/4}$$
where C and B are defined in [19,22]. When the temperature rises above circa 1/2 of the Debye temperature ${ϴ}_{D}$, optical phonons of crystal lattice vibrations appear. Optical phonons have higher energy than acoustic phonons, enough to cause negative energy-level differences between atoms, allowing electron jumps (phonon-assisted hopping—PAH) between neighboring centers. Phonon-assisted DC conductivity is thermally activated and follows the Arrhenius-like relation of Mott’s polaron hopping [16,17,18,19,22,23]:(2)$$\sigma \left(T\right)=\frac{\sigma_{pa}}{T}exp\left(-\frac{E_{DC}}{k_{B}T}\right)\Rightarrow ln\sigma \left(T\right)=ln\frac{\sigma_{pa}}{T}-\frac{E_{DC}}{k_{B}}T^{-1}$$
where $E_{DC}$ is the activation energy, $\sigma_{pa}\propto R^{-1}$, and R is the average distance between the hopping centers. A smooth transition from VRH to PAH is described by the Debye temperature ${ϴ}_{D}$, known from specific heat theory. Well below that temperature, phonons are frozen to an extent, and the assistance of phonons in electron hopping is limited.

On the other hand, random or electric-field-driven hopping of ions in solids (Na^+^ in this case) requires empty sites in a close arrangement of atoms, such as vacancies or interstitial positions. Such conditions occur in many crystalline and amorphous structures. Cathode materials must be additionally electrochemically active, which means their ability to intercalate and de-intercalate ions (e.g., Na^+^), ensuring discharging and charging a battery, respectively. Because of its much larger size and smaller mobility, the conductivity of ions in most systems is usually much lower than the conductivity of electrons (exceptions are superionic conductors and ionic crystals). One of the most studied cathode materials is polycrystalline olivines [7,9], alluaudites [24], and NASICONs [25].

The main experimental method used in this work was broadband dielectric spectroscopy (BDS), which allowed for the measurement of complex electric permittivity [26,27,28,29]:(3)$$\varepsilon^{*}\left(\omega \right)=\varepsilon_{\infty }+\frac{\left(\varepsilon_{s}-\varepsilon_{\infty }\right)}{1+{\left(i\omega \tau \right)}^{n}}=\varepsilon^{\prime }\left(\omega \right)+i\varepsilon^{\prime \prime }\left(\omega \right)$$

In this formula, $\varepsilon_{\infty }$ is the high-frequency electric permittivity, $\varepsilon_{s}$ is the static electric permittivity, and n is the empirical power exponent. This equation illustrates how the electric permittivity varies with frequency, incorporating high- and low-frequency limits and various relaxation phenomena.

Electric permittivity (Equation (3)) can be transformed into electric conductivity, considering $\sigma^{*}=i\varepsilon \varepsilon_{0}\omega$ [26]:(4)$$\sigma^{*}\left(\omega \right)=\sigma_{\infty }+\frac{\left(\sigma_{DC}-\sigma_{\infty }\right)}{1+{\left(i\omega \tau \right)}^{n}}=\sigma^{\prime }\left(\omega \right)+i\sigma^{\prime \prime }\left(\omega \right)$$
where $\sigma_{\infty }$ is the high-frequency conductivity (infinite frequency limit), $\sigma_{DC}$ is the DC conductivity (zero frequency limit), and τ is the relaxation time.

Our last paper [18] was devoted to DC conductivity studies for NaFePO_4_ glasses and resulted in nanomaterials obtained after high-pressure and high-temperature treatment (HPHT). It was found that after HPHT treatment, the glassy sample transformed into a composite consisting of two nanocrystalline electrochemically active phases: 55% alluaudite Na_2_Fe_3_(PO_4_)_3_ and 45% NASICON Na_3_Fe_2_(PO_4_)_3_. The average grain sizes were 42 nm and 91 nm, respectively [18]. The studies using the broadband dielectric spectroscopy (BDS) method showed that DC conductivity increased by 1–2 orders of magnitude after HPHT treatment compared with NaFePO_4_ glasses.

While DC conductivity is a valid phenomenological parameter describing battery operation, AC conductivity analysis may give a more in-depth picture of transport processes occurring in the studied material. For empirically describing electric conductivity changes, Jonscher’s scaling law is most often used [27]:(5)$$\sigma^{\prime }(\omega )=\sigma_{DC}+A\omega^{n}$$

This power law (also called universal dielectric response) shows that the real part of complex conductivity consists of two components: a constant term $\sigma_{DC}$ representing the conductivity at very low frequencies (DC limit) and a frequency-dependent term $A\omega^{n}$ that accounts for the increase in conductivity with increasing angular frequency ω=2πf. In the above formula, A and n are fitting parameters. The power exponent 0<n≤1 determines the rate at which the conductivity increases with frequency. It is assumed that both parts of $\sigma^{\prime }$ are thermally activated and may be expressed as follows [30,31,32,33,34]:(6)$$\sigma_{DC}=\sigma_{0}exp\left(-\frac{E_{DC}}{k_{B}T}\right)$$
(7)$$A=A_{0}exp\left(-\frac{E_{AC}}{k_{B}T}\right)$$
where $E_{DC}$ and $E_{AC}$ are activation energies of DC and AC conductivity, respectively. There are several models postulating the following empirical relationship between both activation energies [30,31,32,33,34,35,36,37,38]:(8)$$E_{AC}=(1-n)E_{DC}$$

The authors’ previous report [18] focused on the effect of high-pressure and high-temperature treatment (HPHT) on the structure properties and DC conductivity of NaFePO_4_ glasses. DC conductivity increased by 2 orders of magnitude, and activation energy of electric conductivity decreased from $E_{DC}=0.57\;eV$ to $E_{DC}=0.41\;eV$ after HPHT treatment. Study in AC domain should allow for analysis of the relation between long-range (DC) and short-range (AC) transport of charge and HPHT treatment impact on this relationship.

## 2. Materials and Methods

At the beginning of the study, a glassy analog of sodium olivine with a nominal composition of NaFePO_4_ was fabricated. As detailed earlier, the glasses were prepared using a melt-quenching technique [18]. Stoichiometric quantities of Na_2_CO_3_, FeC_2_O_4_·2H_2_O, and NH_4_H_2_PO_4_ were carefully mixed and finely ground in a mortar. Subsequently, the mixture was placed into a ceramic crucible and heated in an AFI-02 furnace (Argenta, Brzeziny, Poland) electric furnace to 1523 K, keeping this temperature to ensure the completion of the calcination reaction and volatilization of components. Finally, the melts were quenched onto copper plates. The amorphous state of the samples was confirmed by XRD and DTA methods. Results of material measurements (XRD, DTA) and dielectric (BDS) measurements (including DC conductivity and related analysis of relaxation processes) of both studied samples are available in the previous work [18].

Electrical conductivity was measured using broadband dielectric spectroscopy (BDS) employing Novocontrol Alpha-A High Performance Frequency Analyzer (Novocontrol Technologies GmbH & Co. KG, Hundsangen, Germany). Samples were placed between capacitor plates at a distance 0.5 mm, and the voltage of 1 V was applied with frequencies ranging from 10 mHz to 10 MHz. Measurements were performed at temperatures from 123 to 473 K. BDS experimental data were collected and analyzed using Novocontrol WinDeta 5.88 and WinFit 3.4. Further data analysis was performed using OriginPro 2019b (OriginLab Corporation, Northampton, MA, USA).

As detailed in previous work [18], after electrical measurements at atmospheric pressure, the amorphous material underwent high-pressure–high-temperature treatment (HPHT), a crucial procedure in this study. The HPHT method [17,18,19] was developed in IHHP PAS and successfully applied for the densification of crack-resistant glass [21]. HPHT method is unique to IHPP PAS, as it is a simultaneous application of isostatic high pressure (HP) and high temperature (HT) to high volumes of up to 1 L. Each glass sample was placed in a graphite crucible inside a high-pressure chamber (Unipress Equipment, Warsaw, Poland) [16,17,18,19]. Using a graphite heater, this chamber facilitated precise regulation of inert gas (N_2_) pressure and temperature. The sample was subjected to a pressure of 1 GPa and heated up to 973 K (above the crystallization temperature of 901 K) to induce the nano-nucleation. The sample was then held in such an environment for 15 min. Next, the sample was rapidly cooled down to 873 K (between crystallization and glass transition temperatures of 873 K). Maintaining these conditions for over 30 min led to stabilizing the sample properties and their permanent preservation after returning to normal conditions. Finally, the material was cooled to room temperature, and the gas (i.e., nitrogen, pressurized medium) was decompressed to ambient pressure level (a more in-depth explanation is available in [18]). Long-term monitoring confirmed that the effect of HPHT treatment on the sample’s conductivity was stable for months. Values of temperatures and pressure were chosen based on thermal analysis under pressure [17] and previous experiments with a lithium-based material [15,16].

Figure 1 presents the real part of complex conductivity spectra taken for temperatures within (123 K–473 K) range, before and after HPHT treatment. The imaginary component of electric conductivity is shown in Figure A1, and the results for aged samples are presented in Figure A2. Generally, the single spectrum at a given temperature consists of two parts. The first one is a low-frequency plateau, corresponding to $\sigma_{DC}$ and the second one-corresponds to AC conductivity. The onset frequency (border frequency between two parts) shifts into lower frequencies while temperature decreases. One must remember that the total electric conductivity resulting from electronic and ionic components is measured. In cathode materials like olivine-like LiFePO_4_ or NaFePO_4_, electronic components strongly predominate. As shown in Figure 1a, at lower temperatures, the AC parts of the spectra start to dominate, and their slopes increase. Below 200 K, the AC conductivity component covers the whole frequency range, and no DC component is observed. At T=123 K, an empirical relationship between conductivity and frequency may be expressed by $\sigma (f)\propto f^{0.9}$ (see below). One may also observe some perturbation in the mid-frequency range of the spectra, especially noticeable for temperatures from 273 K to 323 K. It is apparently due to different responses of mobile electrons and ions to stimulating AC signal. It is consistent with the electric modulus studies described in ref. [18].

Figure 1b shows a significant increase in the real part of AC conductivity after HTHP treatment. In this case, the middle-frequency parts of the spectra are even more pronounced, and onset frequencies shift to higher values when compared to Figure 1a. Previous work [18] showed that after HPHT treatment, the material consisted of two nanocrystalline, electrochemically active phases: NASICON (55%) and alluaudite (45%). Under such conditions, the discrepancy between ionic and electronic conductivity components is more visible, apparently due to the heterogeneity of the final nanocomposite. One may observe that true DC conductivity can only be measured in a limited temperature range (visible plateau ranging wide frequency range). However, the use of computer software fitting programs allowed for the estimation of apparent DC conductivity for the frequency tending to zero. In this paper, it will be marked with $\sigma_{DC}$.

The temperature dependencies of apparent DC conductivity for the samples before and after HPHT treatment are shown in Figure 2. The pressure-induced increase in $\sigma_{DC}$ is significant and reaches 2 or even 3 orders of magnitude, depending on the temperatures. According to Mott’s theory [22,23] of electron hopping in disordered systems, mentioned in the Introduction, two temperature regimes can be distinguished. The low-temperature one corresponds to variable range hopping, and the higher-temperature regime is related to thermally activated phonon-assisted hopping [18]. The activation energy of phonon-assisted hopping determined by us in [18], for a narrower temperature range and from $\log(\sigma_{DC}T)=F(1/T)$ plot, decreased from 0.57 eV (before HPHT) to 0.41 eV (after HPHT). The corresponding activation energies determined in this study from $\log(\sigma_{DC})=F(1/T)$ empirical plots are equal: 0.55 eV and 0.38 eV, respectively.

It is notable that the total electric conductivity with a strong predominance of electronic components is measured. We postulate that mobile ions cause some irregularities, visible in Figure 2, in the domain 200–300 K.

Ionic transference numbers $t_{i}$ could be estimated from the Arrhenius plot, shown in Figure 2, supplemented by the normalized formula:(9)$$t_{i}=\frac{\sigma_{i}}{\sigma_{i}+\sigma_{e}}$$
where $\sigma_{i}$ and $\sigma_{e}$ are estimated as ionic and electronic conductivity, respectively. The Arrhenius plot of ionic conductivity exhibits greater slope, because the activation energy of ionic conduction is higher than the activation energy for electronic conduction, and ionic conduction contributes to total conductivity mainly in the high-temperature range. In temperature 473 K, the ionic contribution allows for the estimation of the transference numbers of glass and nanocomposite, which are 0.66 and 0.80, respectively. The crossover is linked to the value 0.5, estimating the crossover between discussed mechanisms.

## 3. Results and Discussion

One of the methods showing the universality of Jonscher’s scaling law in Equation (5) is a representation of $\sigma^{\prime }(\omega )$ spectra in the form of a so-called *master curve* [39,40]. Empirical crossover frequency $f^{*}$ between the DC conductivity and AC conductivity is thermally activated with the same activation energy as $\sigma_{DC}T$ and defined as follows [39]:(10)$$\sigma^{\prime }\left(f^{*}\right)=2\sigma_{DC}$$

Considering this, one can plot normalized-conductivity values of $\log(\sigma /\sigma_{DC})$ against normalized frequency $f_{norm}$ [39]:(11)$$f_{norm}=f\sigma_{DC}^{-1}T^{-1}$$

In the model case, all spectra should converge to a single curve with a high-frequency slope equal to one. The idea of master curves was more or less successfully applied to ionic conductive glasses [31,39,40]. Numerous reports also deal with electronic or mixed electronic–ionic conductors, where electric charge transport occurs by a hopping mechanism [41,42,43,44,45]. The results are depicted in Figure 3.

As one can see, the plot shown in Figure 3a is close to the proper master curve except for the middle-frequency range, in which we postulate the overlapping of electronic and ionic hopping effects. This overlapping is even more pronounced in Figure 3b, which relates to AC conductivity after HPHT treatment. Interestingly, after HPHT treatment, the ordering of single-resolved curves in this range is inverted compared to the situation before treatment. (conf. Figure 3a,b).

Figure 4 presents the temperature behavior of the exponent n from Jonscher’s scaling law (Equation (5)) for the samples before and after HPHT (high-pressure and high-temperature treatment). The values of exponent n change with temperature from 0.3 to 0.9. This change signifies a gradual transition of the frequency-dependence of electric conductivity. A value of n=0.3 (or lower) typically indicates a case where the conductivity increases with frequency at a slower rate, known as sublinear behavior. On the other hand, when n=1, the conductivity exhibits a linear relationship with frequency, implying that the log of conductivity increases proportionally with the frequency logarithm. Therefore, the change observed in Figure 4 reflects a variation from linear to sublinear frequency-dependence in the conductivity behavior on the log–log scale. Two temperature ranges of n variations for the glassy state correlate well with two different transport regimes described earlier (variable-range hopping and phonon-assisted hopping). It is worth noting that values of n almost linearly decrease in the first range and stabilize around n≈0.4 in the second one.

An equivalent 3D network of resistors (R) and capacitors (C) often describes the response of solids subjected to the AC field. It is assumed that power exponent n in Jonscher’s scaling law (Equation (5)) is related to the fraction of resistors and capacitors in that network [46,47]. It is postulated that for values of n lower than 0.5, predominance of resistors takes place, and consequently, transport of charge carriers occurs mainly through percolation paths consisting of resistors (for n=0, there is only DC conduction). On the other hand, for n higher than 0.5, capacitors predominate, and percolation paths consist mainly of those elements (for n=1, there is pure AC conduction and $\sigma_{AC}\propto \omega$). The value of n=0.5 is, therefore, a border between long-range migration and local movements of charge carriers.

The empirical run of exponent n after HPHT treatment is more complicated (Figure 4). Because DC conductivity could only be estimated, only an effective, averaged value of the exponent could be estimated ($n_{eff}$). Up to 200 K behavior is almost the same as in glassy state. When phonon-assisted hopping begins, the value of $n_{eff}$ decreases more rapidly than the glass case. The most intriguing observation relates to the temperature range between 270 and 320 K, when the value of $n_{eff}$ does not decrease but increases from around 0.45 to 0.55 (around the border value). That might mean a transient return to capacitor predominance in transport phenomena because of grain boundary effects in the nanocomposite. Furthermore, this evolution may relate to the shift from variable range hopping (VRH) to phonon-assisted hopping and the impact of interactions between sodium ions (Na^+^) and polarons.

Figure 5a presents temperature dependencies of parameter A, describing this part of conductivity, which corresponds to the AC component (Equation (4)). The lower curve in Figure 5 relates to NaFePO_4_ glass and the upper one to nanocomposite (after HPHT treatment). Depending on the temperature, a considerable increase in A is observed after pressure treatment (1–3 orders of magnitude). In the case of the native glass, one can also observe two distinct regions—corresponding to thermally activated phonon-assisted hopping (higher temperatures) and variable range hopping (VRH, lower temperatures). The activation energy of A determined from the high-temperature range is equal to $E_{AC}=0.40\;\mathrm{eV}$. This value is in reasonable agreement with the value of 0.37 eV obtained from an empirical relationship (Equation (8)) [33,34,35,36,37,38], where the value of n≈0.4 (Figure 4) and $E_{DC}\approx 0.57\;\mathrm{eV}$ [18]. As one might expect, $E_{AC}$ is lower than $E_{DC}$. After HPHT treatment, the temperature dependence of A becomes much more complicated because of the presence of two nanocrystalline phases—alluaudite and NASICON [18]. At temperatures above VRH regime (250 K), one may observe a phonon-assisted-hopping regime with $E_{AC}=0.18\;\mathrm{eV}$, which is in good agreement with the calculated value, 0.20 eV, assuming values n=0.5 (Figure 4) and $E_{DC}\approx 0.41\;\mathrm{eV}$ [18]. As one can see, the dependence of A does not exhibit smooth regularity in the 330–500 K temperature range. This may be due to the heterogeneity of the composite (two nanophases) and different responses of electrons and ions to AC signal. It seems that at higher temperatures, the transport of Na^+^ ions begins to dominate. The effective activation energy of that process is high because of a ramified network of grain boundaries and the small mobility of ions. In the low-temperature range, below 200 K, for both cases (before and after HTHP), the model of variable range hopping (VRH) may be fitted (Equation (1)). In Figure 5b, parameter A is plotted against $T^{-0.25}$ where one may observe linear dependence that confirms VRH mechanism [48]. A brief analysis of the effect observed in this figure indicates an increase in the density of electron states (DOS) on the Fermi level after HPHT treatment [49].

The hopping of the Na^+^ ions accompanies the hopping of electrons, but as our recent paper [18] showed, the frequency of ion jumps is at least 1 order of magnitude lower than electron jumps. Perturbation of electron transport by mobile Na^+^ ions is more pronounced after high-pressure treatment. Generally, the high pressure destroys the smooth temperature dependencies of $n_{eff}$ and A due to the build-up of sample heterogeneity (glass → nanocomposite). This is especially visible in the transition range from the first to the second mechanism of the electron-hopping model.

## 4. Conclusions

Despite measurements performed to frequencies as low as 10 mHz, DC conductivity could not be determined. The fitting allowed for the estimation of the apparent DC conductivity. While low-temperature (apparent) DC conductivity should conform to Mott’s law if governed by VRH, neither the reference glass nor nanocomposite samples follow this relation.

The frequency dependencies of AC conductivity in the studied samples fulfill Jonscher’s scaling law (universal dispersion response) (Equation (5)). Depending on the temperature, the values of power exponent n (or $n_{eff}$) vary in the 0.3–0.9 range, and parameter A, describing the AC component of conductivity, is thermally activated. The temperature dependence of A for NaFePO_4_ glass resembles the temperature behavior of DC conductivity with two temperature regimes: variable range hopping (for lower temperatures) and thermally activated phonon-assisted hopping (for higher temperatures). The activation energy of AC conduction determined for glass and nanocomposite are equal to $E_{AC}\approx 0.40\;\mathrm{eV}$ and $E_{AC}\approx 0.18\;\mathrm{eV}$, respectively. They are lower than the corresponding DC values $E_{DC}\approx 0.55\;\mathrm{eV}$ and $E_{DC}\approx 0.38\;\mathrm{eV}$, fulfilling the empirical rule $E_{AC}=(1-n)E_{DC}$. The natural consequence of AC and DC conduction corresponding to local and long-range charge transport, respectively, is shown in Equation (7).

After high-pressure treatment, the DC conductivity cannot be directly determined, even at those low frequencies. However, the fitting allows for the estimation of the apparent DC conductivity. The AC and apparent DC conductivity values increase by 2–3 orders of magnitude, depending on temperature, due to decreased distance between the Fe^2+^/Fe^3+^ hopping centers. The change in effective power exponent $n_{eff}$ from higher values (0.9–0.5 range) to lower ones (stabilized at about 0.4) observed at c.a. 270 K is related to a smooth transition from variable range hopping (VRH) to phonon-assisted hopping (PAH).

To summarize, electron (polaron) conductivity governs glass material’s conductivity. However, the increase in the nanocomposite’s pressure-induced electron conductivity is partially limited by ion-transport reduction due to introduced heterogeneity and grain boundaries limiting long-range ion transport.

## Figures and Tables

**Figure 1 nanomaterials-14-01492-f001:**
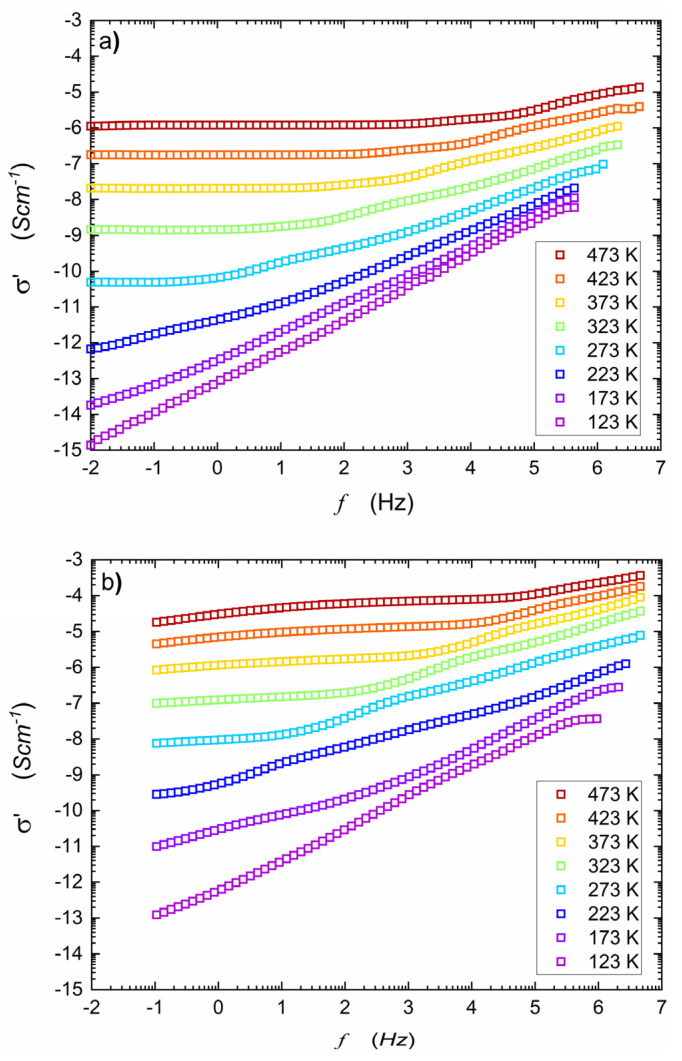
Series of spectra of real parts of the complex conductivity of NaFePO_4_ measured at various temperatures before (**a**) and after (**b**) HPHT treatment (P=1 GPa, T=973 K). A low-frequency *plateau* of DC conductivity seems to emerge in the ‘native’ glass material at high temperatures. It can suggest that the canonical DC electric conductivity in the tested material is absent.

**Figure 2 nanomaterials-14-01492-f002:**
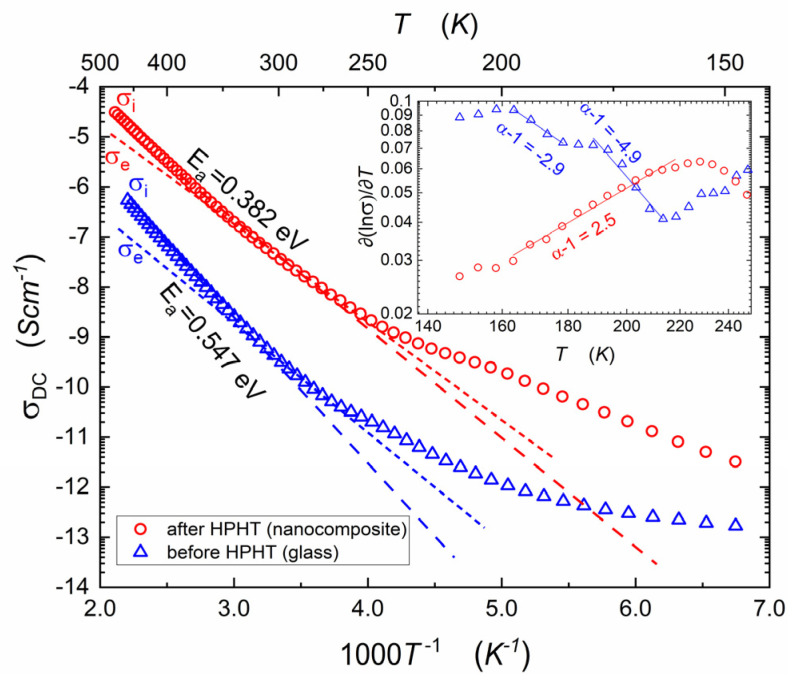
Apparent DC conductivity as a function of temperature for the sample before (glass, blue triangles) and after (nanocomposite, red circles) HPHT treatment. The activation energy (compared with [15]) for the linear regions (marked with dashed lines) is calculated using Equation (6). In the low-temperatures range, the nonlinear behavior, which can be related to the VRH Equation (1a) appears and is indicated by a discrepancy in the dashed lines. The estimated ionic ($\sigma_{i}$) and electronic ($\sigma_{e}$) conductivities are marked. The inset shows a distortion-sensitive test for Mott’s law: derivative of the natural logarithm of conductivity. In case of VRH (described by Mott’s law), this derivate should be linear (in log–log scale), with onset equal to α−1=−0.25−1=−1.25 (comp. Equation (1b)). However, both the only-glass and the nanocomposite do not conform to this relation. The ‘native’ glass behavior does show irregular behavior, with onset locally changing from −2.9 to −4.9, which corresponds to exponent from Equation (1a) α=−1.9 and α=−3.9, respectively. In the case of the nanocomposite, the onset is equal to 2.5, corresponding to the unexpected opposite relationship (and α=3.5).

**Figure 3 nanomaterials-14-01492-f003:**
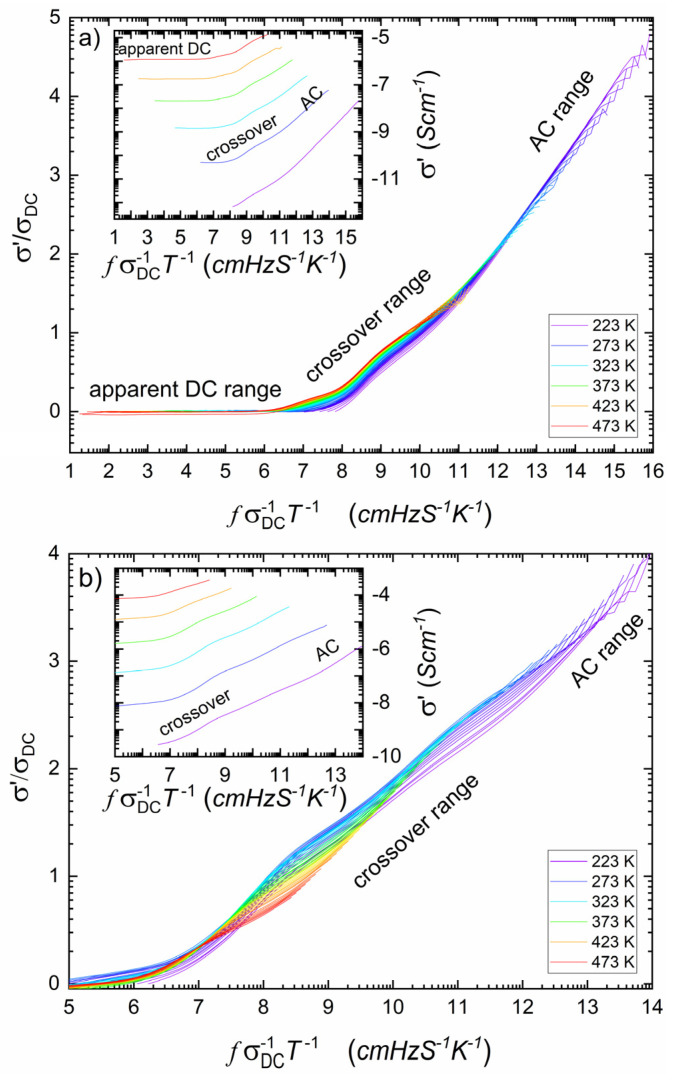
Conductivity normalized to the (apparent) DC conductivity, plotted against the frequency normalized to temperature and the (apparent) DC-conductivity (Equation (10)) measured at different temperatures for NaFePO_4_ before (**a**) and after (**b**) HPHT treatment. Typical convergence can be observed in the glassy material (**a**). In the case of the nanocomposite (**b**), the value of $\sigma_{DC}$ can only be estimated. One may observe the effect of the relaxation processes on medium-range frequencies (crossover range). Insets show $\sigma^{\prime }$ (before $\sigma_{DC}$ scaling) plotted against normalized frequency at a few selected temperatures; one may observe the high-frequency (AC range) onset change and crossover frequency convergence.

**Figure 4 nanomaterials-14-01492-f004:**
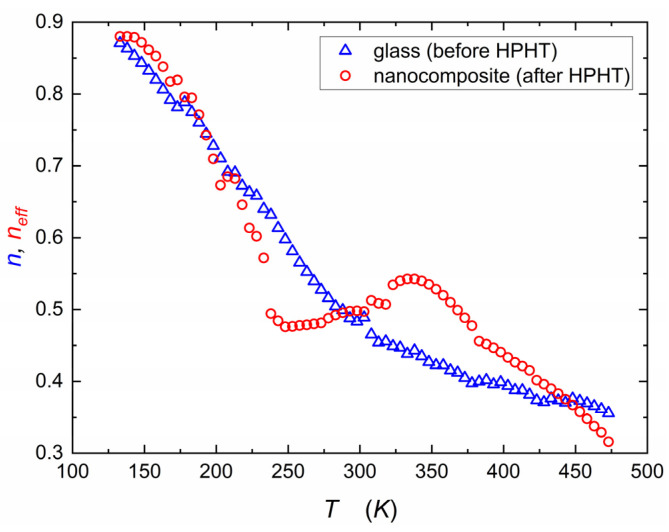
Temperature dependencies of exponent n (Equation (5)) describing linearity of AC conductivity in relation to frequency. With increasing temperature, a gradual decrease can be observed for the glassy material (blue triangles). The nanocomposite (red circles) exhibits three distinct regions: rapid decrease (similar to glassy material) up to 250 K, oscillation close to boundary value of 0.5 (250 K–350 K), and then further decrease (above 350 K). As shown in Figure 3, the nanocomposite’s $\sigma_{DC}$ is only estimated; the exponent $n_{eff}$ is an averaged, effective value.

**Figure 5 nanomaterials-14-01492-f005:**
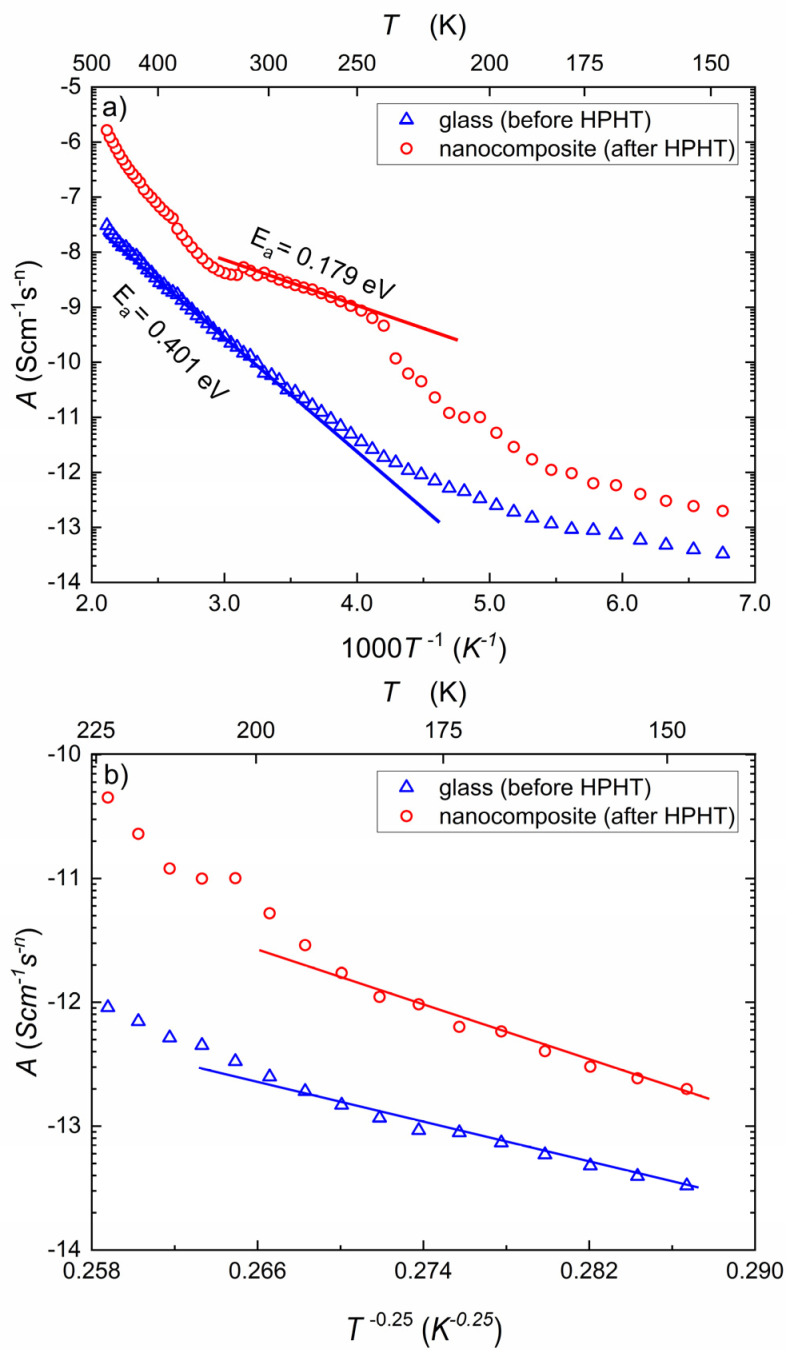
The temperature dependencies of fit parameter A, which govern the relation between temperature and AC conductivity, as fitted from Equation (5). Presented in (**a**) the Arrhenius representation 1/T and (**b**) in the Mott’s representation $1/T^{0.25}$. In the Arrhenius representation (**a**), the AC activation energies of the linear regions are shown (with guidelines), and they correlate to phonon-assisted hopping (Equation (7)). In the Mott’s representation (**b**), the linear regions are highlighted with guidelines, and they correlate to VRH (Equation (1a)). Glass—blue triangles; nanocomposite (after HPHT)—red circles.

## Data Availability

Data are contained within the article.

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
