# Peer review of "AC Electric Conductivity of High Pressure and High Temperature Formed NaFePO4 Glassy Nanocomposite"

_nanomaterials, 2024, doi:10.3390/nano14181492_

Round 1

Reviewer 1 Report

Comments and Suggestions for Authors

Aleksander et al. investigated the effect of high pressure on the DC electric conductivity of NaFePO4 nanocomposites glasses and discussed their prior findings. In this work, the results of AC measurements of glasses and NaFePO4 nanocomposites after pressure treatment were investigated. AC conductivity for glasses and pressure-formed nanocomposites was found to comply with Jonscher's scaling rule, with power exponents ranging from 0.3 to 0.9 depending on temperature. The temperature dependence of AC conductivity was investigated for nanocomposites. After high pressure-high temperature treatment of NaFePO4, AC conductivity increased. High-temperature compression below the glass temperature causes a persistent reduction in the distance between the hopping centres of nanocrystallites in a glassy matrix, resulting in an increase in electric conductivity. On the basis of the results and the discussion, the manuscript may be accepted for publication following a minor revision.

1. It would be better if the authors simplified and made it easier to understand the work's content and concept in the abstract.

2.  It would be better if the authors discussed and tabulated glass- (A), non-glass-based materials (B), and their composite (A+B) as cathodes for battery applications.

3. What is “mixed electronic-ionic electric conduction”? It should be discussed.

4. The authors can discuss the transformation of structural changes in nanomaterials under the influence of pressure, contrasting it with other effects such as heat and cryogenic temperatures.

5. The introduction should discuss the significance of HPHT.

6. What is nanocrystalline? How does it distinguish between bulk and polycrystalline? It should be discussed.

7. What is called for n? Please provide the technical name for "n" on page no. 6.

8.  What is RC?

9. Why does the value of 𝑛 decrease more rapidly than the glass case? It should be discussed.

10. Conclusion looks like a result and discussion. Therefore, it should be simplified. 

Author Response

Comments 1: It would be better if the authors simplified and made it easier to understand the work's content and concept in the abstract.
Response 1: Abstract was refactored; see Lines 10-27.

Comments 2: It would be better if the authors discussed and tabulated glass- (A), non-glass-based materials (B), and their composite (A+B) as cathodes for battery applications.
Response 2: While using such material in the future battery is part of the motivation behind this work, extensive comparison of non-glass and glass-based cathodes is beyond the scope of this work, which is focused on pressure treatment impact on AC conductivity of glass and composite material.

Comments 3: What is “mixed electronic-ionic electric conduction”? It should be discussed.
Response 3: A Short explanation has been added. Lines 66-68.

Comments 4: The authors can discuss the transformation of structural changes in nanomaterials under the influence of pressure, contrasting it with other effects such as heat and cryogenic temperatures.
Response 4: Such measurements for this particular material were not yet performed extensively, as the creation of NaFePO4 glass is novel itself; however, it definitely should be topic of future research. A comparison of heat and heat+pressure treatment is discussed extensively in work 5. such as: [16,19-21]. Added remark in lines 62-65.

Comments 5: The introduction should discuss the significance of HPHT.
Response 5: Discussion of significance of HPHT in the introduction was additionally highlighted. Lines 55-65

Comments 6: What is nanocrystalline? How does it distinguish between bulk and polycrystalline? It should be discussed.
Response 6: Nanocrystalline is polycrystalline material with an average grain size below 100 nm. In ‘grain boundary science’ it may refer to low-size grains on the boundary between bulk material, however, in case of this study, it refers to the full volume of the material. Short comment was added, lines 50-51.

Comments 7: What is called for n? Please provide the technical name for "n" on page no. 6.
Response 7: The technical name of “n” is for the exponent in Jonscher’s scaling law; the reference to this equation was added when required in lines 270-271, 284, 292 and 357.

Comments 8: What is RC?
Response 8: RC is a pair of resistor (R) and capacitor (C) elements; the text was slightly modified to avoid misleading. Line 290.

Comments 9: Why does the value of n decrease more rapidly than the glass case? It should be discussed.
Response 9: Decrease of parameter n corresponds to smooth transition from local range to long range conduction mechanism. The faster rate of change in composite compared to glass is probably due to the crystallinity of composite, which enhances the long-range transport of electric charges. The supplementary explanation was shifted from the Conclusions to the Discussion section. Lines 344-350.

Comments 10: Conclusion looks like a result and discussion. Therefore, it should be simplified.
Response 10: Conclusions were simplified. Lines 362-367.

Kind regards,
Aleksander Szpakiewicz-Szatan

Reviewer 2 Report

Comments and Suggestions for Authors

This manuscript has provided a very deep fundamental study on the AC conductivity of NFP cathode materials. Given the scope and quality of this manuscript, I would think it to be a potential publication here. However, my comments below must be carefully resolved.

1. Why was NaFePO4 chosen as the material of focus, and what are the specific advantages over other potential sodium battery materials? The authors should list a table to do the comparison somewhere in a proper position in the manuscript.

2.  How were these specific conditions (1 GPa and 973 K) determined as optimal?

3. How reproducible are the AC and DC conductivity measurements under the reported experimental conditions? The identical cell results should be included in the Supporting Information with error bar and deviation. 

4. The transition from variable range hopping (VRH) to phonon-assisted hopping is central to the conductivity changes observed. Can you provide a more detailed mechanistic explanation of this transition?

5. The manuscript mentions the disturbance of electronic conductivity by Na+ ions. Can you quantify the ionic contribution and discuss its impact on overall conductivity?

6. As the authors mentioned "Modern energy storage is mainly based on lithium batteries". The following manuscript regarding that regard should be mentioned here also: ACS Applied Materials & Interfaces, 15 (1), 751-760, 2023.

7. Some figures (e.g., Figures 3 and 4) show complex relationships that are not fully explained in the text. Can you provide clearer descriptions or alternative visualizations to aid reader understanding?

Author Response

Comments 1: Why was NaFePO4 chosen as the material of focus, and what are the specific advantages over other potential sodium battery materials? The authors should list a table to do the comparison somewhere in a proper position in the manuscript.
Response 1: A short explanation was added. Lines 40-43.

Comments 2: How were these specific conditions (1 GPa and 973 K) determined as optimal?
Response 2 A short explanation was added. Lines 174-176.

Comments 3: How reproducible are the AC and DC conductivity measurements under the reported experimental conditions? The identical cell results should be included in the Supporting Information with error bar and deviation.
Response 3: Comment in text (lines 179-180) and Figure A2 in Appendix were added to address the reproducibility of results. Lines 404-407.

Comments 4: The transition from variable range hopping (VRH) to phonon-assisted hopping is central to the conductivity changes observed. Can you provide a more detailed mechanistic explanation of this transition?
Response 4: A short explanation was added. Lines 82-84.

Comments 5: The manuscript mentions the disturbance of electronic conductivity by Na+ ions. Can you quantify the ionic contribution and discuss its impact on overall conductivity?
Response 5: The estimated ionic transference number was added with the explanation. Lines 221-222 and 233-241.

Comments 6: As the authors mentioned "Modern energy storage is mainly based on lithium batteries". The following manuscript regarding that regard should be mentioned here also: ACS Applied Materials & Interfaces, 15 (1), 751-760, 2023.
Response 6: Citations have been added in the Introduction as [3]. Lines 32 and 415-417.

Comments 7: Some figures (e.g., Figures 3 and 4) show complex relationships that are not fully explained in the text. Can you provide clearer descriptions or alternative visualizations to aid reader understanding?
Response 7: Additional visual aid has been added for Figure 3 (lines 260-268), and additional comment in text was added for Figure 4 (lines 283-289 and 300-302).

Kind regards,
Aleksander Szpakiewicz-Szatan